# Enhancing Unsupervised Visible-Infrared Person Re-Identification with Bidirectional-Consistency Gradual Matching

## ABSTRACT

Unsupervised visible-infrared person re-identification (USL-VI-ReID) is of great research and practical significance yet remains challenging due to significant modality discrepancy and lack of annotations. Many existing approaches utilize variants of bipartite graph global matching algorithms to address this issue, aiming to establish cross-modality correspondences. However, these methods may encounter mismatches due to significant modality gaps and limited model representation. To mitigate this, we propose a simple yet effective framework for USL-VI-ReID, which gradually establishes associations between different modalities. To measure the confidence whether samples from different modalities belong to the same identity, we introduce a bidirectional-consistency criterion, which not only considers direct relationships between samples from different modalities but also incorporates potential hard negative samples from the same modality. Additionally, we propose a cross-modality correlation preserving module to enhance the semantic representation of the model by maintaining consistency in correlations across modalities. Extensive experiments conducted on the public SYSU-MM01 and RegDB datasets demonstrate the superiority of our method over existing USL-VI-ReID approaches across various settings, despite its simplicity. Our code will be released.

## CCS CONCEPTS

• **Computing methodologies** → **Image representations**; **Visual content-based indexing and retrieval**; *Object identification*; **Matching**.

## KEYWORDS

Unsupervised Visible Infrared Person Re-Identification, Curriculum Learning, Cross-Modality Correlation

## 1 INTRODUCTION

Visible-infrared person re-identification (VI-ReID) aims to retrieve the same person captured by the visible camera when the query image from infrared camera is provided, and vice versa [16, 34, 44]. It has garnered attention for its potential in night vision applications, where traditional visible camera-based methods cannot work well due to poor lighting. Recently, many methods have been proposed for VI-ReID and made impressive progress [1, 17, 25, 32, 48]. However, the requirement for extensive annotations poses a significant challenge and impedes the scalability of these approaches. In response to this limitation, unsupervised visible-infrared person re-identification (USL-VI-ReID) has been raised and become a hot research topic [36, 38].

Due to the absence of precise manual annotations, establishing a robust association between different modalities is crucial in USL-VI-ReID. Recently, many approaches [5, 6, 30, 36] have endeavored to address this challenge by employing variants of bipartite graph matching algorithms. Specifically, these methods usually treat cross-modality label assignment as a graph matching task, aiming to identify correspondences between modalities by minimizing global matching costs [36]. Then, the modality-shareable feature representations can be derived from the generated cross-modality correspondences. However, the global matching approach may lead to mismatches of unreliable samples due to significant modality disparities and the limited representational capacity of the model. This mismatching phenomenon can adversely impact the overall performance of the model. Thus, how to build the reliable association between different modalities remains an open challenging problem nowadays.

In this paper, we rethink the problem of cross-modality label assignments from the perspective of curriculum learning. Regarding the initial limitations of the model in addressing significant modal discrepancies, we advocate for the gradual establishment of associations between different modalities. Initially, we focus on generating cross-modality correspondences for easier samples. As training progresses and the model is more powerful to handle modal discrepancies, we incrementally incorporate more correspondences for harder samples. To measure the confidence whether samples from different modalities belong to the same identity, we propose a simple bidirectional-consistency criteria. This criterion not only leverages the direct relationship between samples from different modalities, but also considers potential hard negative samples from the same modality. However, merely aligning positive pairs from different modalities may weaken the structure information, i.e., the correlations of samples with unmatched clusters from the other modality. To address the concern, we propose the cross-modality correlation preserving module to further enrich the semantic representation of the model by maintaining consistency in correlations across different modalities.

The main contributions of our work can be summarized as follows:

*ACM MM, 2024, Melbourne, Australia*
© 2024 Copyright held by the owner/author(s). Publication rights licensed to ACM.
ACM ISBN 978-x-xxxx-xxxx-x/YY/MM
https://doi.org/10.1145/nnnnnnn.nnnnnnn

- Revisiting the challenge of cross-label assignments through the lens of curriculum learning, we propose a simple yet effective framework for USL-VI-ReID employing a progressive matching paradigm.
- We introduce a straightforward bidirectional-consistency criterion to evaluate whether samples from different modalities share the same identity. This criterion accounts for both positive instances across different modalities and potential negative instances within the same modality.
- To enhance the structure information and the semantic representations of the model, we propose the cross-modality correlation preserving module. This module ensures consistency in correlations across different modalities, thereby enhancing the overall model performance.
- Extensive experiments are conducted on the public SYSU-MM01 and RegDB datasets. The results demonstrate that our proposed method can outperform existing USL-VI-ReID methods under various settings in spite of its simplicity.

## 2 RELATED WORK

### 2.1 Supervised Visible-Infrared Person ReID

Supervised visible-infrared person ReID (SVI-ReID) aims to retrieve the same person across visible and infrared camera views [16, 34, 44]. The key of SVI-ReID is to learn modality-shareable feature representations with accurate manual annotations. Recently, many works have been proposed for SVI-ReID by relieving the discrepancy between different modalities [1, 17, 25, 32, 48]. Among them, [25] proposes to suppress the modality-related features by aligning persons in the pixel-level based on cross-modality dense correspondences. CoAL [32] proposes two novel attention modules to learn discriminative features for each modality and collaborative features across the modality. LUPI [1] generates the intermediate domain between visible and infrared modalities, subsequently mitigating modality shift by incorporating the synthesized intermediary domain as supplementary information. FMCNet [48] aims to generate the missing modality-specific discriminative features for each modality from the information obtained from the other modality, thereby combining these generated features to achieve more precise cross-modality retrieval. [17] proposes a novel data augmentation technique, named PartMix, which can generate positive and negative samples by combing local patches from the images of the same identity and different identities. In addition, some methods aim to reduce the modality gap by employing GANs [10, 26, 33, 40] to transfer the style of the image from one modality to another.

Although these methods have achieved great performance in retrieving the same person across different modalities, they all require expensive manual annotations within the modality and across modalities, which heavily hinders their applications in the real world.

### 2.2 Unsupervised Visible-Infrared Person ReID

Unsupervised visible-infrared person ReID (USL-VI-ReID) aims to match images of the same person across visible and infrared modalities without annotations [22, 38]. Compared with unsupervised single-modality person ReID [3, 20, 28, 46, 50], USL-VI-ReID is more challenging as the modality discrepancy is usually more serious

than the inter-class variance within each modality. To build reliable association between different modalities, many works [5, 6, 30, 36] have been proposed to learn modality-shareable feature representations by generating cross-modality correspondences. Among them, PGM [36] proposes a novel paradigm, which formulates the cross-modality correspondences mining as the graph matching problem. By adopting this approach, the model can effectively incorporate global information through minimizing the global matching cost. OTLA [30] proposes the optimal-transport method to reduce the modality gap by assigning pseudo labels for images from visible modality to infrared modality. To guarantee that all clusters from infrared modality can be matched in the training process, [5] proposes a many-to-many bilateral cross-modality cluster matching method to generate cross-modality correspondences. In addition, [6] proposes a dual optimal transport label assignment method to solve the cross-modality correspondences assignment problem in a reinforcement manner.

Although these methods have achieved great progress on the USL-VI-ReID task with their carefully generated cross-modality correspondences, they usually solve the problem with the global matching algorithms. In this way, some unreliable samples can be mismatched due to serious discrepancy between different modalities and limited feature representations in the early period. To tackle this and avoid label noise accumulation, we propose a novel paradigm to generate correspondences in a gradual manner. To weight the confidence whether samples from different modalities belong to the same identity, we devise a simple yet effective bidirectional-consistency criteria.

### 2.3 Curriculum Learning

Inspired by the human learning process, curriculum learning is proposed by [2] as a training strategy by learning from easy samples to hard samples. Curriculum learning can enhance the performance of the machine learning models, and it has been widely used in many areas, including object detection, segmentation and retrieval [19, 31, 47, 49, 51]. Specifically, [47] proposes a self-paced curriculum learning framework for weakly supervised object detection by utilizing instance-level and image-level prior-knowledge. [49] proposes an adaptive alignment module for remote sensing cross-modal text–image retrieval by utilizing image-text pairs from easy to hard for better feature representations. [31] proposes a selective training framework for the task of node classification by relieving the influence of low-quality training nodes. To prevent the additional bias introduced by human interventions, [51] introduces the curriculum learning strategy in the task of pre-training GNN task by supervising the training process with different structures and feature spaces. To refine the pseudo labels for semi-supervised semantic segmentation, [19] proposes a strategy for confidence score by adopting curriculum learning. To the best of our knowledge, this is the first work to explore cross-modality correspondences generation with curriculum learning.

## 3 METHOD

The framework of our method is illustrated in Fig. 1. In Sec. 3.1, we first introduce the augmented dual-contrastive learning (DCL [38]), which is also regarded as the baseline in our work by learning

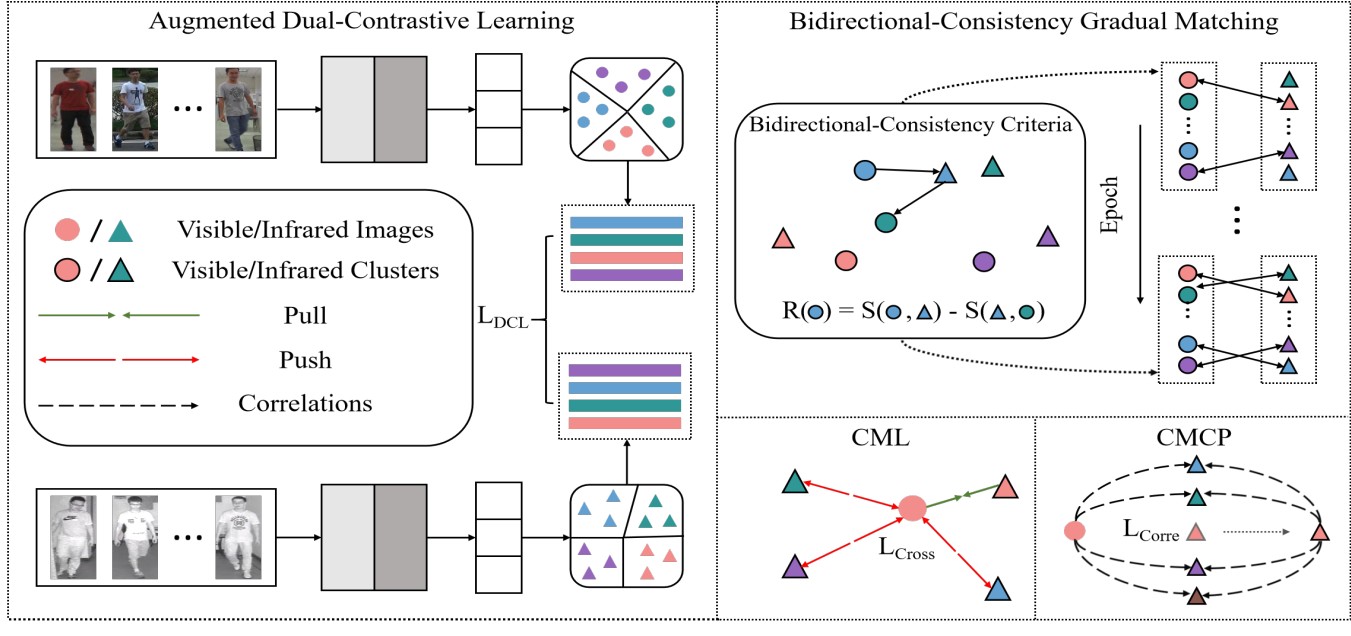

**Figure 1: Framework of the proposed method. Based on Augmented Dual-Contrastive Learning (DCL), we propose the Bidirectional-Consistency Gradual Matching module to generate cross-modality correspondences in a gradual manner. Then Cross-Modality Learning (CML) and Cross-Modality Correlation Preserving (CMCP) modules are applied on the selected samples and their correspondences.**

within each modality. Based on DCL, we further introduce our devised simple bidirectional-criteria and gradual matching strategy, which are described in Sec. 3.2 in detail. Finally, we describe our proposed cross-modality correlation preserving module in Sec. 3.3.

## 3.1 Augmented Dual-Contrastive learning

Given a unlabeled visible and infrared training set $X = \{X_v, X_i\}$, where $X_v = \{x_1^v, x_2^v, \ldots, x_N^v\}$ denotes the visible training set containing $N$ images, while $X_i = \{x_1^i, x_2^i, \ldots, x_M^i\}$ denotes the infrared training set with $M$ images. It is noted that to relieve the discrepancy between different modalities, random Channel Augmentation (CA) [41] is also applied in the visible stream for assistance.

To extract features from images, the two-stream encoders $f_\theta^v$ and $f_\theta^i$ are adopted, which share the same convolution backbone but modality-specific classifiers. In this way, visible feature vectors $U_v = \{u_1^v, u_2^v, \ldots, u_N^v\}$ and infrared feature vectors $U_i = \{u_1^i, u_2^i, \ldots, u_M^i\}$ can be extracted from them with encoders $f_\theta^v$ and $f_\theta^i$, respectively. To generate modality-specific pseudo labels for these training sets, DBSCAN [9] clustering algorithm is applied on $U_v$ and $U_i$ to obtain the corresponding cluster sets $\mathcal{H}_v = \{C_1^v, C_2^v, \ldots, C_L^v\}$ and $\mathcal{H}_i = \{C_1^i, C_2^i, \ldots, C_K^i\}$, where $L$ and $K$ represent the number of clusters for visible and infrared modalities. Then modality-specific prototypes $\Phi_v \in \mathbb{R}^{L \times d}$ and $\Phi_i \in \mathbb{R}^{K \times d}$ can be obtained by averaging the feature vectors within the same cluster, where $d$ is the dimension of the feature vector.

During training, the PK sampling strategy [15] is adopted to construct the mini-batch. For the infrared modality, we randomly sample $P$ identities and each identity contains $K$ images. As the

random channel augmentation is also applied in the visible modality, to balance the number of images from different modalities, we random sample $P$ identities from the visible modality and each identity contains $K/2$ images and $K/2$ corresponding augmented images. Then for the infrared modality, the ClusterNCE [8] loss can be applied as follows,

$$L_{q_i} = -\log \frac{\exp\left(q_i \cdot \phi_+^i / \tau\right)}{\sum_{k=1}^K \exp\left(q_i \cdot \phi_k^i / \tau\right)}, \tag{1}$$

where $q_i$ represents the query sample from infrared modality, $\phi_k^i$ denotes the prototype of the $k$-th cluster from infrared modality and $\phi_+^i$ denotes the prototype of the cluster which $q_i$ belongs to, $\tau$ is the temperature hyper-parameter. Similarly, the ClusterNCE loss can also be applied on the visible and random channel augmented images in the same way by comparing their images with the corresponding prototypes from visible modality, which can be denoted as $L_{q_v}$. In this way, the final objective function of DCL can be obtained by combining them together as follows,

$$L_{DCL} = \frac{1}{|B_I|} \sum_{q_i \in B_I} L_{q_i} + \frac{1}{|B_V|} \sum_{q_v \in B_V} L_{q_v}. \tag{2}$$

where $B_I$ and $B_V$ are input batch from infrared modality and visible modality, respectively. It is noted that $B_V$ also contains samples augmented by random channel augmentation. For simplicity we omit the notation of augmented samples as visible and augmented samples are all included in the same stream and are treated equally. During training, the modality-specific prototypes from visible and infrared modalities can be updated in the momentum manner as

follows,

$$\phi_+^e = m\phi_+^e + (1-m)q_e, \qquad (3)$$

where $q_e$ is the query sample from modality $e$ ($e = \{i, v\}$, representing the infrared modality and visible modality, respectively), $\phi_+^e$ is the prototype of the cluster which $q_e$ belongs from modality $e$. $m$ is the momentum factor, which decides the updating speed of the corresponding prototypes.

Although DCL can relieve the modality discrepancy between visible and infrared modalities to some extent with augmented samples serving as the intermediate connection, it is still insufficient for the model to learn modality-invariant feature representations as it only performs ClusterNCE loss within each modality independently.

## 3.2 Bidirectional-Consistency Gradual Matching

Due to the absence of annotations, the key of USL-VI-ReID lies in establishing the reliable association across different modalities. Different from existing works that establish correspondences by minimizing the overall graph matching cost, our approach advocates for a gradual association construction. Thus, how to design a criteria which can reflect the confidence whether image pairs from different modalities belong to the same identity is the key in our work.

Compared with the unsupervised single-modality ReID, USL-VI-ReID is more challenging as the modality gap is more serious than the inter-class variance within each modality. Directly modeling the similarity score is inadequate for reflecting reliability, as nearby clusters within the same modality may also serve as potential candidates for matching corresponding clusters in other modalities. To capture the structural essence of these clusters, we introduce a simple yet effective bidirectional-consistency criteria. To streamline computational efficiency, we formulate this criterion directly on modality-specific prototypes $\Phi_v \in \mathbb{R}^{L \times d}$ and $\Phi_i \in \mathbb{R}^{K \times d}$. Specifically, given the $\phi_l^v$, which represents the prototype of the $l$-th cluster in the visible modality, we can find its nearest prototype $N^{V \rightarrow I}(\phi_l^v)$ from infrared modality as follows,

$$N^{V \rightarrow I}(\phi_l^v) = \underset{\phi_k^i \in S(I)}{\arg\max}(\phi_l^v \cdot \phi_k^i), \qquad (4)$$

where $S(I)$ denotes the set of prototypes in the infrared modality. Then the bidirectional-consistency criteria can be formed as follows,

$$R^{V \rightarrow I}(\phi_l^v) = \underbrace{\phi_l^v \cdot N^{V \rightarrow I}(\phi_l^v)}_{\text{direct positive similarity}} - \underbrace{\max_{\phi_j^v \in \{S(V)/\phi_l^v\}}\left(N^{V \rightarrow I}(\phi_l^v) \cdot \phi_j^v\right)}_{\text{potential negative similarity}},$$

$$(5)$$

where $R^{V \rightarrow I}(\phi_l^v)$ denotes the similarity between $\phi_l^v$ and its nearest cluster from the other modality, which can reflect the reliability whether they belong to the same identity. $S(V)/\phi_l^v$ denotes the set of prototypes in the visible modality except for $\phi_l^v$. In this way, we can take both the positive samples from different modalities and those potential negative samples from the same modality into fully consideration. By applying Eq. (4), we can obtain the reliability of all $L$ samples in the visible modality as $\mathcal{R}^{V \rightarrow I} = \{R^{V \rightarrow I}(\phi_1^v), R^{V \rightarrow I}(\phi_2^v), \ldots, R^{V \rightarrow I}(\phi_L^v)\}$. Regarding that

the model is limited in dealing with the serious modality discrepancy in the early period, simply apply global alignment like existing works [5, 6, 30, 36] to generate correspondences for all samples may cause the label noise accumulation during training. To build reliable association between different modalities, we aim to generate correspondences in a gradual scheme, i.e., we initially generate correspondences for those most reliable image pairs, then involve harder samples in the matching process when the model is more powerful in the cross-modality representation. Specifically, we firstly sort the reliability set $\mathcal{R}^{V \rightarrow I}$ in the descending order, then select the first $T$-percent samples from it as $\hat{\mathcal{R}}^{V \rightarrow I}$, where the value of $T$ increases linearly with current epoch as follows,

$$T = T_0 + \frac{current\_epoch}{total\_epoch}(1 - T_0), \qquad (6)$$

where $T_0$ is the hyper-parameter that controls the initial proportion of samples to be matched in the beginning, and it is simply set to 0.1 in our experiments. Then, we can obtain the cross-modality correspondences for a subset of samples from the visible modality, which can be represented as follows,

$$\mathcal{M}^{V \rightarrow I} = \left\{(\phi_j^v, N^{V \rightarrow I}(\phi_j^v)) \,\middle|\, \phi_j^v \in \hat{\mathcal{R}}^{V \rightarrow I}\right\}, \qquad (7)$$

where $\mathcal{M}^{V \rightarrow I}$ is the obtained correspondence set for visible modality. Then we can learn cross-modality feature representations with it. Specifically, given the feature vector $q_v$ from the $j$-th cluster in visible modality, if its corresponding prototype $\phi_j^v$ is in the set $\mathcal{M}^{V \rightarrow I}$, the unidirectional learning from visible to infrared can be represented as follows,

$$L_{q_v}^{V \rightarrow I} = -\log \frac{\exp\left(q_v \cdot N^{V \rightarrow I}(\phi_j^v)/\tau\right)}{\sum_{k=1}^K \exp\left(q_v \cdot \phi_k^i/\tau\right)}, \qquad (8)$$

where $N^{V \rightarrow I}(\phi_j^v)$ is the matched prototype in the infrared modality of $\phi_j^v$. By comparing visible images with prototypes in the infrared modality, the modality discrepancy can be efficiently reduced. Similarly, for $q_i$ which is extracted from infrared modality, its unidirectional learning from infrared to visible modality can be obtained in the same way, which is denoted as $L_{q_i}^{I \rightarrow V}$. Then the cross learning can be obtained by combining these two streams as follows,

$$L_{Cross} = \frac{1}{|B_I|} \sum_{q_i \in B_I} L_{q_i}^{I \rightarrow V} + \frac{1}{|B_V|} \sum_{q_v \in B_V} L_{q_v}^{V \rightarrow I}. \qquad (9)$$

In this way, the model can learn cross-modality feature representations in a gradual manner to avoid the label noise accumulation in the early period.

## 3.3 Cross-Modality Correlation Preserving

As shown in Fig. 1, for those selected image pairs from different modalities, cross-modality learning can reduce the modality gap by pulling them together. However, this process may compromising the underlying structure information, particularly the correlation between the sample and other unmatched clusters in the alternate modality by pushing them away. To address this issue, we introduce the cross-modality correlation preserving module, which aim to preserve the underlying structure information for those selected

samples when aligning them with their correspondences. Specifically, considering the feature vector $q_v$ from visible modality, if it is matched with the $l$-th cluster in the infrared modality, then the cross-modality correlation between $q_v$ and all clusters in the infrared modality except for the $l$-th cluster can be represented as the probabilistic vector $P_{q_v}^{V \to I}$. Then, the $l$-th element of $P_{q_v}^{V \to I}$ is assigned a value of 0, while the remaining elements are defined as:

$$\left[ P_{q_v}^{V \to I} \right]_j = \frac{\exp\left( q_v \cdot \phi_j^i / \tau \right)}{\sum_{k=1}^{K} \exp\left( q_v \cdot \phi_k^i / \tau \right) - \exp\left( q_v \cdot \phi_l^i / \tau \right)}, \quad (10)$$

where $\left[ P_{q_v}^{V \to I} \right]_j$ denotes the $j$-th entry of $P_{q_v}^{V \to I}$, $\phi_l^i$ is the correspondence of $q_v$, i.e., the prototype in the infrared modality which is matched with $q_v$. In this way, we can represent the correlation between $q_v$ and all clusters in infrared modality except for the matched $l$-th cluster. Analogously, the intrinsic correlation between the $l$-th cluster with other clusters in the same infrared modality can be represented as $P_{\phi_l^i}^{I \to I}$. Similarly, we set the $l$-th entry of $P_{\phi_l^i}^{I \to I}$ to 0, with the remaining elements defined as follows,

$$\left[ P_{\phi_l^i}^{I \to I} \right]_j = \frac{\exp\left( \phi_l^i \cdot \phi_j^i / \tau \right)}{\sum_{k=1}^{K} \exp\left( \phi_l^i \cdot \phi_k^i / \tau \right) - \exp\left( \phi_l^i \cdot \phi_l^i / \tau \right)}. \quad (11)$$

Given that $q_v$ and $\phi_l^i$ are linked in the matching procedure, they are likely to represent the same identity across different modalities. Therefore, to preserve the cross-modality correlations, we align these probabilistic vectors as follows,

$$L_{corre}^{V \to I}(q_v) = \| P_{q_v}^{V \to I} - P_{\phi_l^i}^{I \to I} \|_2^2. \quad (12)$$

Similarly, for $q_i$ which is extracted from infrared modality, the cross-modality correlation preserving loss can be obtained in the same way by comparing it with visible prototypes, which can be denoted as $L_{corre}^{I \to V}(q_i)$. Then the final cross-modality correlation preserving module can be formed by combing these two terms as follows,

$$L_{Corre} = \frac{1}{|B_I|} \sum_{q_i \in B_I} L_{corre}^{I \to V}(q_i) + \frac{1}{|B_V|} \sum_{q_v \in B_V} L_{corre}^{V \to I}(q_v). \quad (13)$$

**Discussion.** For those matched clusters from different modalities by our bidirectional-consistency gradual matching module, Eq. (9) can align them together by pulling the sample from one modality and the matched cluster from the other modality closer. However, this alignment strategy risks compromising the underlying structure information since Eq. (9) also pushes the sample further away from other unmatched clusters. To address this challenge, we propose the cross-modality correlation preserving module, which aims to preserve the correlation between the sample and clusters from the other modality except for the matched cluster. As the probability vector used in Eq. (13) contains no information about the matched cluster in the other modality, it doesn't violate the goal of Eq. (9) in the training process.

## 3.4 Optimization

Following existing works [5, 6, 36, 38], our model is also trained with two stages. In the first stage, the model is optimized with the DCL module as follows,

$$\mathcal{L}_{stage1} = L_{DCL}. \quad (14)$$

After the first training stage, the model can relieve the modality discrepancy to some extent by leveraging the augmented visible images as intermediate associations. Then the model can be trained by combining our proposed bidirectional-consistency gradual matching module and cross-modality correlation preserving module to further boost the discrimination of its cross-modality feature representations as follows,

$$\mathcal{L}_{stage2} = L_{DCL} + \lambda_1 L_{Cross} + \lambda_2 L_{Corre}, \quad (15)$$

where $\lambda_1$ and $\lambda_2$ are hyper-parameters which balance basic DCL module and our proposed modules.

## 4 EXPERIMENT

### 4.1 Datasets and Evaluation Protocols

Our experiments are conducted on two publicly available VI-ReID datasets: RegDB [24] and SYSU-MM01 [34]. Following existing works [6, 36, 38], we adopt mean average precision (mAP) and Cumulative Matching Characteristics (CMC) as the evaluation metrics. Among the CMC evaluation metric, Rank-1, Rank-10 and Rank-20 are also reported for more exhaustive evaluations. In addition, the mean Inverse Negative Penalty (mINP) metric proposed in [43] is also presented.

SYSU-MM01 is a large-scale public VI-ReID dataset collected from 4 visible cameras and 2 near-infrared cameras. The training set contains 395 identities, including 22,258 visible images and 11,909 infrared images, while the test set contains 96 identities. Following [36], we evaluate our method on this dataset with two settings, including All-search mode and Indoor-search mode. For all-search mode, the gallery set contains images collected from all visible cameras. For indoor-search mode, only images captured by indoor visible cameras are used to constitute the gallery set.

RegDB is collected from one visible camera and one infrared camera. The dataset contains 4,120 visible images and 4,120 infrared images of 412 identities. Our method is evaluated on this dataset with two settings, including visible-to-infrared search and infrared-to-visible search. For fair comparison, following existing works [6, 36, 38], we conduct experiments on this dataset for 10 times and report the average result as the final performance.

### 4.2 Implementation Details

Our method is implemented in the PyTorch platform. Following [36] we adopt the ImageNet-pretrained ResNet50 [14] network as the backbone in the experiment. In the mini-batch, the number of identities for each modality is set to 16 and each identity contains 16 instances. Following [36], we resize the input image as $288 \times 144$. For data augmentation, random horizontal flipping, padding, random cropping, and random erasing, random grayscale and channel argumentation [41] are applied on the input image.

In the training process, DBSCAN [9] clustering algorithm is applied to generate pseudo labels within each modality. For DBSCAN, the minimal number for neighbours is set to 4 for different datasets, the maximum distance is set to 0.6 for SYSU-MM01 dataset while 0.3 for RegDB dataset, which are the same as existing works [36, 38].

**Table 1: Experimental results (%) of the proposed method and SOTA methods on the SYSU-MM01 and RegDB datasets under different settings. ∗ means the model is pre-trained on an extra labeled visible dataset, † means the model is pre-trained on AGW [43] while ‡ means the model is pre-trained on CLIP [27].**

| | Method | Reference | \multicolumn{6}{}{SYSU-MM01 dataset} | | | | | | \multicolumn{6}{}{RegDB dataset} | | | | | |
|---|---|---|---|---|---|---|---|---|---|---|---|---|---|---|
| | | | \multicolumn{3}{}{All Search} | | | \multicolumn{3}{}{Indoor search} | | | \multicolumn{3}{}{Infrared to Visible} | | | \multicolumn{3}{}{Visible to Infrared} | | |
| | | | r1 | mAP | mINP | r1 | mAP | mINP | r1 | mAP | mINP | r1 | mAP | mINP |
| SVI-ReID | DDAG | ECCV-20 | 54.8 | 53.0 | 39.6 | 61.0 | 68.0 | 62.2 | 68.1 | 61.8 | 48.6 | 69.3 | 63.5 | 49.2 |
| | AGW | TPAMI-21 | 47.5 | 47.7 | 35.3 | 54.2 | 63.0 | 59.2 | 70.5 | 65.9 | 51.2 | 70.1 | 66.4 | 50.2 |
| | CA | ICCV-21 | 69.9 | 66.9 | 53.6 | 76.3 | 80.4 | 76.8 | 84.8 | 77.8 | 61.6 | 85.0 | 79.1 | 65.3 |
| | MCLNet | ICCV-21 | 65.4 | 62.0 | 47.4 | 72.6 | 76.6 | 72.1 | 75.9 | 69.5 | 52.6 | 80.3 | 73.1 | 57.4 |
| | MPANet | CVPR-21 | 70.6 | 68.2 | - | 76.7 | 81.0 | - | 82.8 | 80.7 | - | 83.7 | 80.9 | - |
| | MAUM | CVPR-22 | 71.7 | 68.8 | - | 77.0 | 81.9 | - | 87.0 | 84.3 | - | 87.9 | 85.1 | - |
| | DART | CVPR-22 | 68.7 | 66.3 | 53.3 | 72.5 | 78.2 | 74.9 | 82.0 | 73.8 | 56.7 | 83.6 | 75.7 | 60.6 |
| | CTFT | ECCV-22 | 74.1 | 74.8 | - | 81.8 | 85.6 | - | 90.3 | 90.8 | - | 92.0 | 92.0 | - |
| | MUN | ICCV-23 | 76.2 | 73.8 | - | 79.4 | 82.1 | - | 91.9 | 85.0 | - | 95.2 | 87.2 | - |
| | PartMix | CVPR-23 | 77.8 | 74.6 | - | 81.5 | 84.4 | - | 84.9 | 82.5 | - | 85.7 | 82.3 | - |
| USL-ReID | SPCL | NIPS-20 | 18.4 | 19.4 | 11.0 | 26.8 | 36.4 | 33.1 | 11.7 | 13.6 | 10.1 | 13.6 | 14.9 | 10.4 |
| | MMT | ICLR-20 | 21.5 | 21.5 | 11.5 | 22.8 | 31.5 | 27.7 | 24.4 | 25.6 | 18.7 | 25.7 | 26.5 | 19.6 |
| | ICE | ICCV-21 | 20.5 | 20.4 | 10.2 | 29.8 | 38.4 | 34.3 | 12.2 | 14.8 | 10.6 | 13.0 | 15.6 | 11.9 |
| | IICS | CVPR-21 | 14.4 | 15.7 | 8.4 | 15.9 | 24.9 | 22.2 | 9.1 | 9.9 | - | 9.2 | 9.9 | - |
| | CCL | ACCV-22 | 20.2 | 22.0 | 13.0 | 23.3 | 34.0 | 30.9 | 11.1 | 13.0 | 9.0 | 11.8 | 13.9 | 9.9 |
| | ISE | CVPR-22 | 20.0 | 18.9 | 8.5 | 14.2 | 24.6 | 21.7 | 10.8 | 13.7 | 10.7 | 16.1 | 17.0 | 13.2 |
| | PPLR | CVPR-22 | 12.0 | 12.3 | 5.0 | 12.7 | 20.8 | 17.6 | 8.1 | 9.1 | 5.7 | 8.9 | 11.1 | 7.9 |
| USL-VI-ReID | H2H†∗ | TIP-21 | 30.2 | 29.4 | - | - | - | - | - | - | - | 23.8 | 18.9 | - |
| | OTLA | ECCV-22 | 29.9 | 27.1 | - | 29.8 | 38.8 | - | 32.1 | 28.6 | - | 32.9 | 29.7 | - |
| | OTLA∗ | ECCV-22 | 48.2 | 43.9 | - | 47.4 | 56.8 | - | 49.6 | 42.8 | - | 49.9 | 41.8 | - |
| | ADCA | MM-22 | 45.5 | 42.7 | 28.3 | 50.6 | 59.1 | 55.2 | 68.5 | 63.8 | 49.6 | 67.2 | 64.1 | 52.7 |
| | PGM† | CVPR-23 | 57.3 | 51.8 | 35.0 | 56.2 | 62.7 | 58.1 | 69.9 | 65.2 | - | 69.5 | 65.4 | - |
| | CCLNet‡ | MM-23 | 54.0 | 50.2 | - | 56.7 | 65.1 | - | 70.2 | 66.7 | - | 69.9 | 65.5 | - |
| | DOTLA†∗ | MM-23 | 50.4 | 47.4 | 32.4 | 53.5 | 61.7 | 57.4 | 82.9 | 75.0 | 58.6 | 85.6 | 76.7 | 61.6 |
| | MBCCM | MM-23 | 53.1 | 48.2 | 32.4 | 55.2 | 62.0 | 57.1 | 82.8 | 76.7 | 61.7 | 83.8 | 77.9 | 65.0 |
| | Ours | - | 58.9 | 53.6 | 36.5 | 60.3 | **67.0** | **62.8** | 86.1 | 81.0 | 67.7 | 86.5 | 81.8 | 70.1 |
| | Ours† | - | **61.7** | **56.1** | **38.7** | **60.9** | 66.5 | 62.3 | **86.8** | **81.7** | **68.6** | **86.7** | **82.3** | **71.1** |

**Table 2: Ablation study on the SYSU-MM01 and RegDB datasets (%).**

| Index | \multicolumn{5}{}{Components} | | | | | \multicolumn{6}{}{SYSU-MM01 Settings} | | | | | | \multicolumn{6}{}{RegDB Settings} | | | | | |
|---|---|---|---|---|---|---|---|---|---|---|---|---|---|---|---|---|---|---|
| | DCL | BGM | BCR | GM | CMCP | \multicolumn{3}{}{All search} | | | \multicolumn{3}{}{Indoor Search} | | | \multicolumn{3}{}{Infrared-to-Visible} | | | \multicolumn{3}{}{Visible-to-Infrared} | | |
| | | | | | | r1 | mAP | mINP | r1 | mAP | mINP | r1 | mAP | mINP | r1 | mAP | mINP |
| 1 | ✓ | ✓ | | | | 50.1 | 46.2 | 30.1 | 56.4 | 63.1 | 58.6 | 73.6 | 67.4 | 51.9 | 73.6 | 68.5 | 55.3 |
| 2 | ✓ | | ✓ | | | 53.6 | 48.9 | 32.5 | 57.3 | 63.9 | 59.3 | 79.5 | 73.9 | 59.3 | 78.7 | 73.9 | 61.1 |
| 3 | ✓ | ✓ | | ✓ | | 55.0 | 49.2 | 31.6 | 57.2 | 64.2 | 59.9 | 86.1 | 80.7 | 67.0 | 85.7 | 81.0 | 69.3 |
| 4 | ✓ | | ✓ | ✓ | | 57.5 | 51.5 | 33.9 | 58.8 | 65.9 | 61.5 | 85.9 | 80.8 | 67.5 | 86.0 | 81.3 | 69.8 |
| 5 | ✓ | | ✓ | ✓ | ✓ | **58.9** | **53.6** | **36.5** | **60.3** | **67.0** | **62.8** | **86.1** | **81.0** | **67.7** | **86.5** | **81.8** | **70.1** |

As described in Sec. 3.4, our model is trained with two stages. To train the model, Adam optimizer with weight decay 5e-4 is adopted. For each stage, we set the initial learning rate as 3.5e-4, and reduce it every 20 epochs for a total 50 epochs. In Eq. (15), $\lambda_1$ and $\lambda_2$ are set to 0.3 and 0.5, respectively. Following [36], the alternate learning strategy is applied on the cross-modality learning with different directions.

## 4.3 Comparison with State-of-the-art Methods

To demonstrate the effectiveness of our method, we compare our method with state-of-the-art supervised visible-infrared person ReID (VI-ReID), unsupervised single-modality person ReID (USL-ReID) and unsupervised visible-infrared person ReID (USL-VI-ReID) methods on two public datasets under different settings. The results are shown in Tab. 1.

*4.3.1  Comparison with SVI-ReID methods.* We compare our method with existing SVI-ReID methods, including DDAG [42], AGW [43], CA [41], MCLNet [13], MPANet [35], DART [39], MAUM [23], CTFT [21], MUN [45] and PartMix [18]. Although these methods train their models with ground truth, our proposed method can still outperform some of them on these datasets. Specifically, on the SYSU-MM01 dataset, we can achieve competitive performance compared with some methods, such as DDAG and AGW. While on the RegDB dataset, our method can outperform most of them, including DDAG, AGW, CA, MCLNet, MPANet and DART. The reason could be that compared with RegDB dataset, SYSU-MM01 is more challenging and it is limited to train the model with generated pseudo labels.

*4.3.2  Comparison with USL-ReID methods.* We also compare our method with existing USL-ReID methods, including SPCL [12], MMT [11], ICE [3], IICS [37], CCL [8], ISE [50] and PPLR [7]. Although these methods can achieve great performance on the task of unsupervised single-modality person ReID, they are very limited on the USL-VI-ReID task compared with our method. The reason could be that compared with the USL-ReID, in USL-VI-ReID the modality discrepancy is usually more serious than the inter-class variance within each modality. Thus, it is hard to directly apply the clustering algorithms to generate cross-modality correspondences for the task of USL-Vi-ReID, which verifies the necessity in generating correspondences between different modalities.

*4.3.3  Comparison with USL-VI-ReID methods.* We compare our method with existing USL-VI-ReID methods, including H2H [22], OTLA [30], ADCA [38], PGM [36], CCLNet [4], DOTLA [6] and MBCCM [5]. Among them, H2H, OLTA and DOTLA utilize the extra labeled visible dataset while CCLNet utilize the pretrained CLIP as the encoder. Compared with these methods, our method can achieve better performance on both SYSU-MM01 and RegDB datasets under different settings. The reason could be that these methods build the association between different modality with global matching algorithms. In this way, some unreliable samples can be mismatched due to the serious modality discrepancy and limited feature representations of the model in the early period. Different from them, we generate cross-modality correspondences in a gradual manner, which can relieve the label noise accumulation when associating different modalities.

## 4.4  Ablation Study

In this section, we conduct experiments on SYSU-MM01 and RegDB datasets to evaluate the effectiveness of different components in our method, and the results are shown in Tab. 2. The definitions of different components are explained as follows: DCL is the baseline in our work, which is described in Sec. 3.1. BGM (bipartite graph matching) is a common matching algorithm used by existing works [5, 6, 36], it aims to associate different sample by minimizing the global matching cost. BCR (Bidirectional-Consistency Criteria) and GM (Gradual Matching Strategy) are our proposed key components described in Sec. 3.2 for generating correspondences in a gradual manner. CMCP (Cross-Modality Correlation Preserving) is our proposed component described in Sec. 3.3.

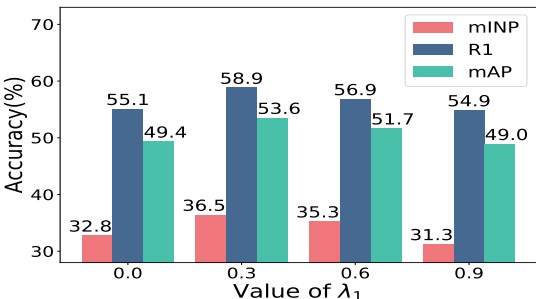

**Figure 2: Impact of hyper-parameter $\lambda_1$ on SYSU-MM01 dataset under all search setting.**

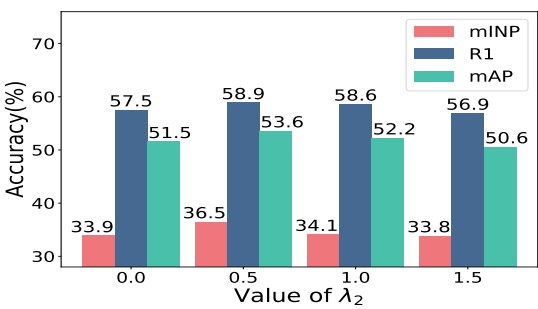

**Figure 3: Impact of hyper-parameter $\lambda_2$ on SYSU-MM01 dataset under all search setting.**

*4.4.1  Effectiveness of BCR.* The comparison between Index 1 and Index 2 shows the effectiveness of our proposed BCR module, which surpasses the performance of BGM by 2.7% and 6.5% in terms of mAP on different datasets. Different from BGM which utilizes the bipartite graph matching algorithm to align modalities based on the direct similarity between different samples, our proposed BCR can weight the reliability of the sample and their matched correspondence by taking the potential negative samples into consideration.

*4.4.2  Effectiveness of GM.* The results of Index 2 and Index 4 show that GM module can further boost the performance of our BCR module by generating the correspondences in a gradual manner. Specifically, the GM improves the performance of BCR by 2.6% and 6.9% in terms of mAP on SYSU-MM01 and RegDB datasets. Furthermore, by comparing Index 1 and Index 3 it can be found that GM can also improve the performance of BGM by 3.0% and 3.3% in terms of mAP on SYSU-MM01 and RegDB datasets.

*4.4.3  Effectiveness of CMCP.* The comparison between Index 4 and Index 5 verify the effectiveness of our CMCP module. Specifically, CMCP can further improve the performance of our method by 2.1% and 0.2% in terms of mAP on SYSU-MM01 and RegDB datasets. Compared with SYSU-MM01 dataset, the improvement on RegDB dataset is a bit limited and the reason could be that the combination of BCR and GM has achieved satisfactory performance, it is hard for CMCP to further boost them due to limited scope for further improvement.

## 4.5 Hyper-parameter Analysis

$\lambda_1$ and $\lambda_2$ in Eq. (13) are two important hyper-parameters introduced in our work. In this section, we conduct experiments to evaluate the influence of these hyper-parameters in our work.

*4.5.1 Influence of $\lambda_1$.* In the objective function, $L_{DCL}$ can learn within the modality to promote more accurate clusters for association, while $L_{Cross}$ explores the cross-modality relationships based on the learned homogeneous feature representations. Thus, how to balance them in the learning process is crucial in our work. In Fig. 2, we conduct experiments on SYSU-MM01 dataset to determine the value of $\lambda_1$. The results show that our method is relatively robust against $\lambda_1$. When $\lambda_1$ is set to 0, our method can still achieve good performance, the reason could be that our method can benefit from preserving correlations with CMCP module. When $\lambda_1$ is set to 0.3, our method can achieve the best performance.

*4.5.2 Influence of $\lambda_2$.* $\lambda_2$ determines the weight of our CMCP module in the final objective function. Fig. 3 shows the results under different values of $\lambda_2$ on SYSU-MM01 dataset. As illustrated in the figure, setting $\lambda_2$ to 0.5 and 1.0 yields superior performance compared to the performance when $\lambda_2$ is set to 0, which verifies the effective of our CMCP module. However, when $\lambda_2$ is assigned with a larger value, such as 1.5, the performance of the model shows a notable decline, the reason can be that the overemphasis on $\lambda_2$ weaken the influence of DCL and cross-modality learning modules.

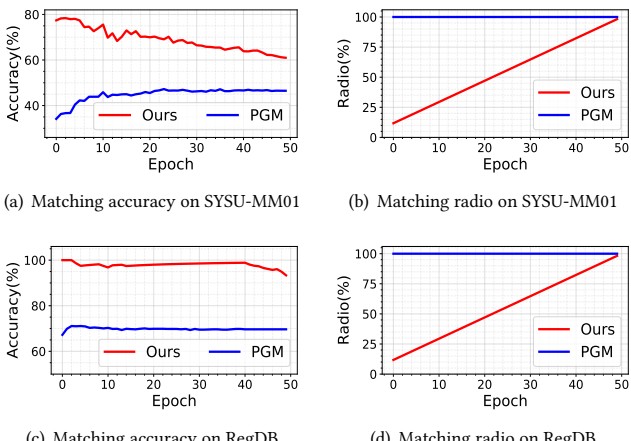

(a) Matching accuracy on SYSU-MM01     (b) Matching radio on SYSU-MM01

(c) Matching accuracy on RegDB     (d) Matching radio on RegDB

**Figure 4: Cross-modality matching accuracy and matching radio of PGM and our method.**

## 4.6 Visualization

*4.6.1 Evaluating the quality of generated correspondences.* In the task of USL-VI-ReID, cross-modality correspondences are necessary for associating different modalities due to the absence of annotations. Different from existing works [5, 6, 36] that typically employ variants of bipartite graph matching algorithms to establish cross-modality correspondences (i.e., determining whether samples from different modalities have the same identity), our approach advocates for a gradual correspondence discovery process to mitigate

mismatches stemming from global alignment issues. In Fig. 4, we evaluate the cross-modality matching accuracy and matching radio in PGM [36] and our method. As depicted in the figure, the matching accuracy of our method significantly surpasses the accuracy of PGM. The reason could be that the model learned simple patterns at first, enabling it to handle more complex matching rules.

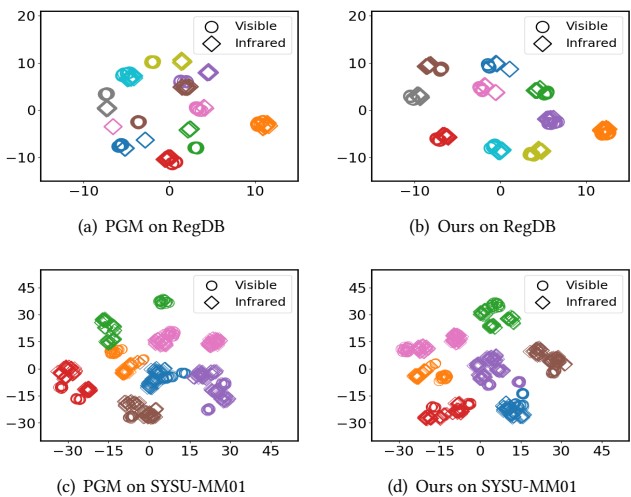

(a) PGM on RegDB     (b) Ours on RegDB

(c) PGM on SYSU-MM01     (d) Ours on SYSU-MM01

**Figure 5: T-SNE visualization of features learned by PGM and our method on a subset of RegDB and SYSU-MM01 datasets. Different colors represent different identities.**

*4.6.2 T-SNE visualization.* To further investigate the effectiveness of our method in learning discriminative feature representations, we employ t-SNE [29] to visualize features learned by PGM and our method, and results are shown in Fig. 5. For RegDB, we randomly select 10 identities, while for another challenging SYSU-MM01 dataset, 7 identities are randomly selected. From Fig. 5 (a) and Fig. 5 (b), we can find that PGM tends to separate features of the same identity across modalities, while our method produces compact clustering of features from different modalities belonging to the same identity. On another challenging SYSU-MM01 dataset (Fig. 5 (c) and Fig. 5 (d)), both PGM and our method show a larger variance among the same identity. However, for some hard samples, e.g., samples with green and orange colors, PGM may confuse them due to incorrect association caused by global alignment during training.

## 5 CONCLUSION

In the paper, we propose a simple yet effective framework for USL-VI-ReID by establishing the association between different modalities through a gradual matching approach. To measure the reliability that samples from distinct modalities belong to the same identity, we introduce a straightforward bidirectional-consistency criteria, which accounts for both intra-modal and inter-modal relationships. Furthermore, we introduce the cross-modality correlation preserving module to maintain the coherence of correlations across modalities. Extensive experiments on the public SYSU-MM01 and RegDB datasets demonstrate the effectiveness of our approach under various settings.

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
