# OpenReview forum: "Enhancing Unsupervised Visible-Infrared Person Re-Identification with Bidirectional-Consistency Gradual Matching"
_acmmm.org/ACMMM/2024/Conference — MM2024 Poster_

### Official Review · Reviewer_zgCP · 2024-04-29

**Rating:** 4
**Confidence:** 4

**Summary:**

This paper investigates the unsupervised VI-ReID task. A Bidirectional Matching algorithm with a gradual matching strategy is proposed to progressively establish reliable cross-modality associations, drawing inspiration from the concept of curriculum learning. Additionally, a Cross-modality Correlation Preserving module is introduced to maintain the structural relationships between instances and their unmatched cross-modality prototypes.

**Strengths:**

(1) The proposed GM strategy addresses cross-modality association through the lens of curriculum learning, distinguishing it from existing methods.\
(2) Comprehensive experiments validate the effectiveness of each component of the method.

**Limitations:**

(1) While BCR introduces a straightforward bi-directional cross-modality association algorithm, this paper specifically compares it with the uni-directional BGM. The efficacy of bi-directional strategies has been demonstrated in prior works [1]. Therefore, it is recommended that the authors include a comparison between BCR and bi-directional methods (e.g., DOTLA [1] and a bi-directional BGM [2]) to validate its superiority. \
(2) The CCP module ensures structural consistency between instances and their matched prototypes, whereas existing self-consistency-based methods [3,4] focus on maintaining instance-level consistency. It is suggested that the advantages of CCP over these methods be elucidated. \
(3) The visualizations in the paper appear hastily done, the authors are recommended to standardize the figures in the experiments.

**Suitability:**

3

---

### Official Review · Reviewer_eJT3 · 2024-05-24

**Rating:** 4
**Confidence:** 4

**Summary:**

This paper aims to address the USL-VI-ReID task and proposes a bidirectional-consistency criterion to measure the confidence whether samples from different modalities belong to the same identity. Besides, this paper also proposes a cross-modality correlation preserving module to enhance the semantic representation of the model by maintaining consistency in correlations across modalities. Extensive experiments conducted on the public SYSU-MM01 and RegDB datasets demonstrate the superiority of the proposed method over existing USL-VI-ReID approaches across various settings.

**Strengths:**

1.The method is relatively simple and effective, and the author clearly explains the proposed method.
2.Extensive experiments have demonstrated the effectiveness of the proposed method.

**Limitations:**

1.In Section 4.3, the author should focus on comparing and analyzing the advantages and disadvantages of USL-VI-ReID based methods.

2.In line 574, "argument" seems to be "authorization".

3. Some newer related methods need to be referenced, including but not limited to the following methods:
[1] Yang B, Chen J, Ye M. Towards grand unified presentation learning for unsupervised visible informed person re identification [C]//Proceedings of the IEEE/CVF International Conference on Computer Vision 2023: 11069-11079
[2] Yang B, Chen J, Chen C, et al. Dual Consistency Constrained Learning for Unsupervised Visible Informed Person Re Identification [J] IEEE Transactions on Information Forensics and Security, 2023.

4. It would be better if the author could provide experimental results on the LLCM dataset, which is a more realistic VI-ReID dataset.
[3] Zhang Y, Wang H. Diverse embedding expansion network and low-light cross-modality benchmark for visible-infrared person re-identification[C]//Proceedings of the IEEE/CVF conference on computer vision and pattern recognition. 2023: 2153-2162.

**Suitability:**

2

---

### Official Review · Reviewer_nnjD · 2024-05-24

**Rating:** 4
**Confidence:** 2

**Summary:**

This paper introduces a USL-VI-ReID framework designed from the perspective of curriculum learning. It introduces bidirectional consistency standards to measure the confidence of whether samples from different modalities belong to the same identity. Additionally, it proposes a cross-modal correlation preservation module aimed at enhancing the semantic representation of the model by maintaining consistency in cross-modal correlations.

**Strengths:**

1. They propose a simple yet effective framework for USL-VI-ReID employing a progressive matching paradigm.
2. They propose the cross-modality correlation preserving module to enhance the model performance and gain SOTA results.

**Limitations:**

1. The author claims that merely aligning positive pairs from different modalities may weaken the structural information. How can this be validated?

2. The motivation for the proposed Augmented Dual-Contrastive learning is not clear in the Introduction.

3. What are the advantages of the proposed Augmented Dual-Contrastive Learning over other methods that use Contrastive Learning? Please provide relevant experimental comparisons and analyses.

4. How to validate that the proposed method establishes more reliable associations between different modalities? Please conduct relevant experiments and analysis.

5. The terminator for Formula (2) should be ',' instead of '.'.

**Suitability:**

2

---

### Official Review · Reviewer_C7Gr · 2024-05-27

**Rating:** 4
**Confidence:** 4

**Summary:**

This paper proposes a bidirectional-consistency gradual matching paradigm for unsupervised visible-infrared person re-identification (USL-VI-ReID), which gradually establishes associations between visible modality and infrared modality. To measure the confidence whether samples from different modalities belong to the same identity, this paper introduces a straightforward bidirectional-consistency criterion, which not only considers positive instances across different modalities but also incorporates potential hard negative instances within the same modality. Additionally, this paper devises a cross-modality correlation preserving module to enhance the structure information and the semantic representations of the model by maintaining consistency in correlations across modalities. Extensive experiments conducted on public SYSU-MM01 and RegDB datasets demonstrate the superiority of the proposed method over existing state-of-the-art approaches across various settings in spite of its simplicity.

**Strengths:**

1.This paper rethinks the problem of cross-modality label assignments from the perspective of curriculum learning and proposes a straightforward and effective strategy to gradually establish cross-modality correspondences, which significantly improves matching accuracy and relieves modality discrepancy.

2.This paper is well written and the experiments are detailed and plausible. An exhaustive ablation study and hyper-parameter analysis are provided to verify the effectiveness and robustness of the proposed method.

3.Compared with the latest USL-VI-ReID approaches in recent years, the proposed method demonstrates superior performance on public SYSUMM01 and RegDB datasets under various settings.

**Limitations:**

1.In Fig. 4, this paper compares the cross-modality matching accuracy and matching radio (typo, which should be “ratio”) of PGM with the proposed method. In Sec. 3.2, this paper has already claimed that the matching ratio grows linearly from $T$-percent to one hundred percent as the current epoch number increases. Therefore, Fig. 4(b) and (d) are redundant. In addition, it is recommended that Fig. 4(a) and (c) be explained and discussed in further detail.

2.In Sec. 3.1, this paper claims that the two-stream modality encoders share the same convolution backbone but modality-specific classifiers. However, the modality-specific classifiers do not appear in Fig. 1.

3.Some equations in this paper are improperly formulated. For example, $\hat{\mathcal{R}}^{V\to I}$ in Eq. (7) denotes the set of the reliability of selected samples in the visible modality while $\phi_j^{v}$ denotes the prototype of the $j$-th cluster in the visible modality. Therefore, $\phi_j^{v} \in \hat{\mathcal{R}}^{V\to I}$ is incorrectly formulated.

4.There are a few typos in this paper. For example, the abbreviation “VI-ReID” for supervised visible-infrared person ReID in Sec. 4.3 should be corrected to “SVI-ReID”. In Sec. 4.5, “$\lambda_{1}$ and $\lambda_{2}$ in Eq. (13)“ should be corrected to “$\lambda_{1}$ and $\lambda_{2}$ in Eq. (15)”.

**Suitability:**

3

---

### Meta-Review · Area_Chair_8A8u · 2024-06-30

**Recommendation:** Accept (Poster)
**Confidence:** 5

**Metareview:**

This paper got 3 WA and 1 BA, and the authors have addressed most of the concerns from the reviewer.